# Granulocyte-Macrophage-Colony-Stimulating-Factor Combined with Prostaglandin E1 Create Dendritic Cells of Leukemic Origin from AML Patients’ Whole Blood and Whole Bone Marrow That Mediate Antileukemic Processes after Mixed Lymphocyte Culture

**DOI:** 10.3390/ijms242417436

**Published:** 2023-12-13

**Authors:** Marianne Unterfrauner, Hazal Aslan Rejeski, Anne Hartz, Sophia Bohlscheid, Tobias Baudrexler, Xiaojia Feng, Elias Rackl, Lin Li, Andreas Rank, Giuliano Filippini Velázquez, Christoph Schmid, Jörg Schmohl, Peter Bojko, Helga Schmetzer

**Affiliations:** 1Department of Medicine III, University Hospital of Munich, 81377 Munich, Germany; 2Department of Hematology and Oncology, University Hospital of Augsburg, 86156 Augsburg, Germany; 3Department of Hematology and Oncology, Diakonieklinikum Stuttgart, 70176 Stuttgart, Germany; 4Department of Hematology and Oncology, Rotkreuzklinikum Munich, 80634 Munich, Germany

**Keywords:** AML, whole blood, whole bone marrow, immune escape, leukemia-derived dendritic cells, dendritic cell-based therapy

## Abstract

Although several (chemotherapeutic) protocols to treat acute myeloid leukemia (AML) are available, high rates of relapses in successfully treated patients occur. Strategies to stabilize remissions are greatly needed. The combination of the (clinically approved) immune-modulatory compounds Granulocyte-Macrophage-Colony-Stimulating-Factor (GM-CSF) and Prostaglandine E1 (PGE-1) (Kit-M) converts myeloid blasts into dendritic cells of leukemic origin (DC_leu_). After stimulation with DC_leu_ ex vivo, leukemia-specific antileukemic immune cells are activated. Therefore, Kit-M treatment may be an attractive immunotherapeutic tool to treat patients with myeloid leukemia. Kit-M-mediated antileukemic effects on whole bone marrow (WBM) were evaluated and compared to whole blood (WB) to evaluate the potential effects of Kit-M on both compartments. WB and WBM samples from 17 AML patients at first diagnosis, in persisting disease and at relapse after allogeneic stem cell transplantation (SCT) were treated in parallel with Kit-M to generate DC/DC_leu_. Untreated samples served as controls. After a mixed lymphocyte culture enriched with patients’ T cells (MLC), the leukemia-specific antileukemic effects were assessed through the degranulation- (CD107a^+^ T cells), the intracellular IFNγ production- and the cytotoxicity fluorolysis assay. Quantification of cell subtypes was performed via flow cytometry. In both WB and WBM significantly higher frequencies of (mature) DC_leu_ were generated without induction of blast proliferation in Kit-M-treated samples compared to control. After MLC with Kit-M-treated vs. not pretreated WB or WBM, frequencies of (leukemia-specific) immunoreactive cells (e.g., non-naive, effector-, memory-, CD3^+^β7^+^ T cells, NK- cells) were (significantly) increased, whereas leukemia-specific regulatory T cells (T_reg_, CD152^+^ T cells) were (significantly) decreased. The cytotoxicity fluorolysis assay showed a significantly improved blast lysis in Kit-M-treated WB and WBM compared to control. A parallel comparison of WB and WBM samples revealed no significant differences in frequencies of DC_leu_, (leukemia-specific) immunoreactive cells and achieved antileukemic processes. Kit-M was shown to have comparable effects on WB and WBM samples regarding the generation of DC_leu_ and activation of (antileukemic) immune cells after MLC. This was true for samples before or after SCT. In summary, a potential Kit-M in vivo treatment could lead to antileukemic effects in WB as well as WBM in vivo and to stabilization of the disease or remission in patients before or after SCT. A clinical trial is currently being planned.

## 1. Introduction

### 1.1. Acute Myeloid Leukemia

Acute myeloid leukemia (AML) is a hematological malignancy defined by uncontrolled expansion of myeloid blasts [1]. According to the hypothesis of the simplified “two-hit model of leukemogenesis”, in many cases, two types of mutations are involved in AML pathogenesis: one resulting in uncontrolled proliferation (e.g., *FLT3*, *NRAS*) and the other leading to impairment of physiological myeloid differentiation (e.g., *TP53*, *CEBPA*) [2]. Clinical symptoms of AML patients are heterogeneous and include hemorrhages and severe infections due to bone marrow failure [3,4].

### 1.2. The Physiologic and Leukemic Bone Marrow Niche

The bone marrow (BM) represents the main site of hematopoiesis [5]. The endosteum with its vessels forms the niche responsible for maintaining a constant pool of hematopoietic stem cells (HSCs) [5,6]. In a dense network of vasculature and fenestrated sinusoids occupying most of the space of BM, HSCs differentiate into erythrocytes, progenitor, myeloid and lymphoid cells. The highly-regulated interplay of reticular cells, endothelial cells, HSCs, mesenchymal stromal cells (MSCs), osteoblasts, adipocytes, immune cells, chemokines, and growth and adhesion factors is instrumental for the controlled differentiation of all blood cell lineages in the steady-state and in response to stress [5,7,8].

In AML however, the physiological BM niche is disrupted and remodeled into an environment favorable for leukemic growth and progression [8,9]. MSCs seem to play an important role in this development since MSCs isolated from AML patients vs. those from healthy donors showed a higher immunosuppressive effect. A dysregulated cytokine production from MSCs and leukemic blasts (e.g., inhibited IFNγ and increased IL-10 production) was shown to lead to impaired T cell functions and further blast proliferation [10,11,12]. In consequence, it was shown that the number of regulatory T cells (T_reg_) was increased, and dendritic cell (DC) and natural killer (NK) cell functions were inhibited [10]. Leukemic blasts were shown to adhere to stromal cells, leading to the persistence of minimal residual disease and chemoresistance [9]. Furthermore, AML cells can change adipocytes’ function to promote their own metabolism. The transport of fatty acids from adipocytes was shown to be induced and thus benefits the proliferation of blasts [13]. Moreover, in leukemia, the number of osteoblasts, important for the physiological differentiation of HSCs, was shown to be decreased [8]. Furthermore, neoangiogenesis in BM might be induced by angiogenic growth factors secreted by blasts and could play a role in the progression of leukemia [9].

### 1.3. Standard Treatment and Advances in AML Therapy

For decades, the standard AML therapy has been the administration of a “7 + 3” chemotherapy regimen (7 days Cytarabine and 3 days Daunorubicin) [2]. Until today, allogeneic hematopoietic stem cell transplantation (HSCT) is the only curative treatment option for most patients [14]. Relapse rates after chemotherapy varying from 30% (younger patients with favorable risk) to 80% (elderly patients with adverse risk) [15], and relapse rates after HSCT of 40% demonstrate that therapies to stabilize remissions are highly needed [16]. Enhanced understanding of the molecular mechanisms behind AML and the potential of blasts to escape the immune system has led to the development of targeted therapies using specific antibodies and cells [17]: e.g., antibodies targeting CD33 (e.g., Gemtuzumab-Ozogamicin), CD123 [17,18,19] or checkpoint molecules (e.g., Nivolumab) [20]. Furthermore, Midostaurin, inhibiting *FLT3* mutations (being present in 30% of newly diagnosed adults) [21], Venetoclax, inhibiting the anti-apoptotic protein BCL2 [22], and Ivosidenib or Enasidenib, inhibiting *IDH1* or *IDH2* mutations (being present in 7–19% of AML patients) [3], have become crucial components in AML therapy. The only approved drug for maintenance therapy (for patients in first remission following chemotherapy) is an oral hypomethylating agent: Azacytidine [23].

Another very promising approach is the use of chimeric antigen receptor (CAR) T cells, genetically modified T cells binding leukemia-specific antigens and inducing antileukemic effects [3,17,24]. Finally, even if the AML blasts might be effectively combated by treatment, leukemic stem cells (LSCs) often persist (in the bone marrow or in the body tissue) through immune escape mechanisms that are not entirely understood and are responsible for relapses [2].

### 1.4. DC/DCleu-Based Therapy

DCs are potent antigen-presenting cells (APCs) bridging the gap between the innate and adaptive immune system and have the potential to express a full spectrum of (infectious or tumor) antigens in a costimulatory manner to immune cells [19,25].

In blood, DC can be subdivided into two major DC subsets: conventional DC (cDC), further divided into cDC1 (main surface markers: CD141^+^, CLEC9A^+^, XCR1^+^) and cDC2 (main surface markers: CD1c^+^ and CD172a^+^), found in blood, lymphoid and non-lymphoid tissue and recognized as the major antigen-presenting cells and plasmacytoid DC (pDC; main surface markers: CD123^+^, CD303^+^ and CD304^+^) secreting type 1 interferon [26,27,28]. DCs have the ability to recognize pathogen- or damage-associated patterns (PAMP or DAMP), subsequently undergo maturation processes (through, e.g., CCR7 expression), migrate to secondary lymphoid organs, present antigen fragments via major histocompatibility complex (MHC) I or II to naive T cells, and activate them through costimulatory molecules (e.g., CD80) and cytokine release [29].

A specialty of (myeloid) leukemic blasts is that they can be directly converted to “leukemia-derived dendritic cells” (DC_leu_) [30]. Their leukemic derivation can be proven by “FISH-IPA”, combining FISH-analysis of chromosomal aberration with surface staining (for DC phenotype) on slides [31] or flow cytometry (codetecting DC and blast markers) [32]. This is the best method to prove leukemic derivation of DC in all patient samples and is regularly used by us [33,34].

These DC_leu_, presenting both DC- and individual patients’ blast antigens, can initiate (patient-specific) anti-leukemic immune responses [30,32]. The comparison of several DC/DC_leu_-generating protocols showed that Kit-M, a combination of Granulocyte-Macrophage-Colony-Stimulating-Factor (GM-CSF) and Prostaglandin E1 (PGE1), had the highest potential to produce DC/DC_leu_ from blast-containing leukemic whole blood (WB) (thereby stimulating physiological conditions) without induction of blast proliferation [33] (thereby simulating physiological conditions in the blood) and to induce anti-leukemic activity in AML patients [35,36]. In contrast to peripheral blood mononuclear cells (PBMCs) previously used to generate DCs, WB contains the broad spectrum of individual patients’ soluble and cellular components [36]. With regard to potential future applications of Kit-M to convert blasts (in the hematopoietic system) to DC_leu_ in vivo, its effect on whole-bone-marrow-(WBM)-blasts in their inhibitory microenvironment and the primary center of leukemic origin needs to be further evaluated.

### 1.5. Aim of the Study

The aim of this study was to explore whether Kit-M treatment could overcome WBM’s inhibitory microenvironment and lead to similar antileukemic effects as already shown in WB. Therefore, experiments with leukemic WB and WBM were performed in parallel, and results were compared.

In detail, we assessed:Composition of immune cells in uncultured (as well as after mixed lymphocyte culture (MLC) with Kit-M-treated vs. untreated) WB and WBMDC (subsets) from Kit-M-treated vs. untreated leukemic WB and WBMPlatelet counts in WB and WBM DC culture supernatants under the influence of Kit-M vs. GM-CSF(Leukemia-specific/antileukemic) immunoreactive cells (activating/inhibitory cells of the adaptive and innate immune system: NK-, CIK-, iNKT- and T-cell subtypes) were specified using degranulation (DEG), intracellular cytokine (INCYT) as well as cytotoxicity (CTX) assays in uncultured WB and WBM and after MLC with Kit-M-treated vs. untreated WB and WBMDifferences of results obtained in WB vs. WBMCorrelation analyses of ex vivo generated data with patients’ allocation to clinical risk groups

## 2. Results

### 2.1. Uncultured WB and WBM

#### 2.1.1. Composition of T and Innate Immune Cells in Uncultured Leukemic WB and WBM

No significant differences in the composition of T and innate immune cells were found in uncultured WB as compared to WBM (Figure 1).

Flow cytometric analyses were performed to quantify T and innate immune cell subtypes in uncultured WB and WBM. Mean frequencies ± standard deviation (SD) of cell subtypes are given. Differences were considered as “highly significant” in cases with *p*-values ≤ 0.005, as “significant” with *p*-values ≤ 0.05, as “borderline significant” with *p*-values between 0.05 and 0.1, and as “not significant” (n.s.) with *p*-values ≥ 0.1. Abbreviations of cell subpopulations are given in table in Section 4.3. 

#### 2.1.2. Composition of (Leukemia-Specific) Degranulating or Intracellularly IFNγ- (and TNFα-) Producing Immunoreactive Cells in Uncultured WB and WBM

In WB (Figure 2A, left side), we found no significant differences in degranulating (CD107a^+^) immunoreactive cells with LAA stimulation (WB^+LAA^) vs. without (WB^−LAA^). We found significantly higher frequencies of intracellularly IFNγ-producing (IFNγ^+^) CD3^+^IFNγ^+^, CD3^+^CD4^+^IFNγ^+^, and CD3^+^β7^+^IFNγ^+^ cells in WB^+LAA^ as compared to WB^−LAA^ (Figure 2B, left side).

Similar results were obtained comparing degranulating WBM^+LAA^ vs. WBM^−LAA^ (Figure 2A, right side). In WBM, we found significantly higher frequencies of CD3^+^CD8^+^IFNγ^+^, central memory (T_cm_) IFNγ^+^, and CD3^+^β7^+^IFNγ^+^, and borderline significantly higher frequencies of CD3^+^IFNγ^+^ cells (Figure 2B, right side) in WBM^+LAA^ as compared to WBM^−LAA^.

We detected significantly lower frequencies of CD4^+^T_reg_CD107a^+^ in WB^−LAA^ as compared to WBM^−LAA^. No significant differences in frequencies of CD107a^+^ cells were found comparing WB^+LAA^ to WBM^+LAA^ (Figure 2A, left side vs. right side).

Significantly higher frequencies of CD3^+^CD8^+^IFNγ^+^ T cells were found comparing WB^−LAA^ with WBM^−LAA^. We detected no significant differences in WB^+LAA^ vs. WBM^+LAA^ (Figure 2B, left side vs. right side).

Similar results were obtained studying intracellularly TNFα-producing immune cells [37].

The degranulation (Figure 2A) and the intracellular cytokine (Figure 2B) assay were performed to detect leukemia-specific cells in uncultured WB and WBM. Cells were stained with CD107a to detect degranulation activity and with IFNγ to examine intracellular cytokine production. Mean frequencies ± standard deviation (SD) of degranulating and IFNγ-secreting T and innate immune cells with or without previous LAA stimulation are given. Differences were considered as “highly significant” in cases with *p*-values ≤ 0.005, as “significant” with *p*-values ≤ 0.05, as “borderline significant” with *p*-values between 0.05 and 0.1, and as “not significant” (n.s.) with *p*-values ≥0.1. Double-sided arrows give (significant) differences between defined immunoreactive cell subtypes in WB and WBM. Abbreviations of cell subpopulations are given in table in Section 4.3. 

### 2.2. Effects of Kit-M on the Generation of Mature DC/DC_leu_ from Leukemic WB and WBM

In WB, we generated significantly higher frequencies of DC, DC_leu_, and mature DC (DC_mat_) after treatment with Kit-M (WB^DC(M)^) compared to control (WB^DC(Control)^) (e.g., %DC_leu_/WB: WB^DC(M)^: 9.59 ± 8.09; WB^DC(Control)^: 5.99 ± 4.08, *p* = 0.04). Blasts’ proliferation was not induced under the influence of Kit M (Figure 3A,B, left side).

In WBM, we found significantly higher frequencies of DC-subtypes comparing WBM^DC(M)^ with WBM^DC(Control)^ (e.g., %DC_leu_/WBM: WBM^DC(M)^: 11.35 ± 9.11; WBM^DC(Control)^: 8.42 ± 7.15, *p* = 0.008). Blasts’ proliferation was not induced under the influence of Kit-M (Figure 3A,B, right side).

A parallel comparison of frequencies of DC-subtypes obtained with WB^DC(Control)^ vs. WBM^DC(Control)^ and WB^DC(M)^ vs. WBM^DC(M)^ (Figure 3A,B, left side vs. right side) revealed no significant differences.

WB and WBM samples were cultured in parallel for 7 days with Kit-M or without added Kit as control. Results with Kit-M (WB^DC(M)^ and WBM^DC(M)^) and without added Kit as control (WB^DC(Control)^ and WBM^DC(Control)^) are given. Mean frequencies ± SD of DC (subtypes) and proliferating blasts from leukemic WB (left side, Figure 3A,B) and leukemic WBM (right side, Figure 3A,B) samples are given. Differences were considered as “highly significant” in cases with *p*-values ≤ 0.005, as “significant” with *p*-values between 0.05 and 0.1, and as “not significant” (n.s.) with *p*-values ≥ 0.1. Abbreviations of cell subpopulations are given in table in Section 4.3.

### 2.3. Effects of Kit-M-Treated WB and WBM on Immunoreactive Cells before and after T Cell-Enriched Mixed Lymphocyte Culture (MLC)

We noticed a generally higher activation status of immune cells in WB after MLC (WB^DC-MLC^) vs. WB^DC^ and in WBM^DC-MLC^ vs. WBM^DC^ due to the stimulating effect of IL-2 [38].

After MLC, we found significantly higher frequencies of early proliferating (T_prol-early_), non-naive (T_non-naive_), effector memory (T_em/eff_) and CD3^+^β7^+^ T cells comparing WB^DC-MLC(M)^ with WB^DC-MLC(Control)^ (e.g., %T_non-naive_/CD3^+^: WB^DC-MLC(M):^ 62.28 ± 22.74; WB^DC-MLC(Control)^: 50.08 ± 25.37, *p* = 0.001). Furthermore, we found a significant decrease in frequencies of CD152^+^ expressing T cells (CD3^+^CD152^+^) in WB^DC-MLC(M)^ vs. WB^DC-MLC(Control)^. Regarding cells of the innate immune system, we found significantly higher frequencies of NK and iNKT cells comparing WB^DC-MLC(M)^ with WB^DC-MLC(Control)^ (e.g., %NK/cells: WB^DC-MLC(M)^: 10.42 ± 9.14; WB^DC-MLC(Control)^: 5.57 ± 4.84, *p* = 0.03). Other cell subsets showed no significant differences in the groups compared (Figure 4, left side).

In WBM, we found significantly higher frequencies of CD3^+^4^+^, CD3^+^CD8^+^, T_prol-early_, T_non-naive_, T_cm_, and CD3^+^β7^+^ T cells comparing WBM^DC-MLC(M)^ with WBM^DC-MLC(Control)^ (e.g., %T_non-naive_/CD3^+^: WBM^DC-MLC(M)^: 58.32 ± 21.67; WBM^DC-MLC(Control)^: 49.22 ± 23.43, *p* = 0.01). We detected significantly lower frequencies of CD3^+^CD8^+^ and CD4^+^T_reg_ in WBM^DC-MLC(M)^ vs. WBM^DC-MLC(Control)^. Significantly higher frequencies of NK and CIK cells were found comparing WBM^DC-MLC(M)^ with WBM^DC-MLC(Control)^ (e.g., %NK/cells: WB^DC-MLC(M)^: 6.08 ± 2.67; WB^DC-MLC(Control)^: 4.08 ± 3.11, *p* = 0.02). No other significant results were obtained from the groups compared (Figure 4, right side).

A parallel comparison of WB and WBM samples revealed significantly lower frequencies of CD3^+^CD152^+^ in WB^DC-MLC(M)^ as compared to WBM^DC-MLC(M)^ (Figure 4, left side vs. right side).

WB^DC^ and WBM^DC^ were enriched with autologous T cells, stimulated with IL-2, and cultured for 7 days in MLC. Given are the mean frequencies ± SD of T cell subsets and of cells of the innate immune system after MLC in WB^DC-MLC(Control)^ and WB^DC-MLC(M)^ (left side) and in WBM^DC-MLC(Control)^ and WBM^DC-MLC(M)^ (right side). Differences were considered as “highly significant” in cases with *p*-values ≤ 0.005, as “significant” with *p*-values ≤ 0.05, as “borderline significant” with *p*-values between 0.05 and 0.1, and as “not significant” (n.s.) with *p*-values ≥ 0.1. Double-sided arrows give (significant) differences between defined cell subtypes in WB and WBM. Abbreviations of cell subpopulations are given in table in Section 4.3. 

### 2.4. Effects of Kit-M-Treated WB and WBM on the Degranulation Activity of Immunoreactive Cells after MLC

In WB, significantly higher frequencies of CD3^+^CD107a^+^, T_non-naive_CD107a^+^, T_em/eff_CD107a^+^, and T_cm_CD107a^+^ were found comparing WB^DC-MLC(M)^ with WB^DC-MLC(Control)^ (e.g., %T_non-naive_CD107a^+^/T_non-naive_: WB^DC-MLC(M)^: 54.54 ± 24.65; WB^DC-MLC(Control)^: 47.32 ± 22.55, *p* = 0.002). We showed a significant decrease in CD4^+^T_reg_CD107a^+^ in WB^DC-MLC(M)^ vs. WB^DC-MLC(Control)^ (Figure 5, left side). 

In WBM, we obtained similar results to WB comparing WBM^DC-MLC(M)^ to WBM^DC-MLC(Control)^ (e.g., %T_non-naive_CD107a^+^/T_non-naive_: WBM^DC-MLC(M)^: 69.70 ± 27.06; WBM^DC-MLC(Control)^: 53.88 ± 24.68, *p* = 0.02). However, we could not detect significant differences of CD4^+^T_reg_CD107a^+^ in the groups compared (Figure 5, right side).

A parallel comparison of frequencies obtained from WB^DC-MLC^ vs. WBM^DC-MLC^ showed no significant differences (Figure 5, left side vs. right side).

Kit-M-pretreated WB^DC-MLC(M)^ and WBM^DC-MLC(M)^ and untreated WB^DC-MLC(Control)^ and WBM^DC-MLC(Control)^ were stained with an antibody against CD107a to detect degranulating T cells as a marker for induced cell cytotoxicity. Given are the mean frequencies ± SD of T cell subsets expressing CD107a in leukemic WB (left side) and leukemic WBM (right side) without LAA stimulation. Differences were considered as “highly significant” in cases with *p*-values ≤ 0.005, as “significant” with *p*-values ≤ 0.05, as “borderline significant” with *p*-values between 0.05 and 0.1, and as “not significant” (n.s.) with *p*-values ≥ 0.1. Abbreviations of cell subpopulations are given in table in Section 4.3.

### 2.5. Effects of Kit-M-Treated WB and WBM on the Intracellular IFNγ (and TNFα) Production of Immunoreactive Cells after MLC

In WB, we found significantly higher frequencies of CD3^+^IFNγ^+^, CD3^+^CD8^+^IFNγ^+^, CD3^+^β7^+^IFNγ^+^, T_non-naive_IFNγ^+^, T_em/eff_IFNγ^+^ and T_cm_IFNγ^+^ comparing WB^DC-MLC(M)^ with WB^DC-MLC(Control^) (e.g., %CD3^+^IFNγ^+^/CD3^+^: WB^DC-MLC(M)^: 43.79 ± 18.77; WB^DC-MLC(Control)^: 34.99 ± 18.05, *p* = 0.00002) (Figure 6, left side). 

In WBM, higher frequencies of almost all previously mentioned cells were identified comparing WBM^DC-MLC(M)^ with WBM^DC-MLC(Control)^, though only CD3^+^IFNγ^+^ and CD3^+^β7^+^IFNγ^+^ showed significant differences in the 7 parallelly studied cases (e.g., %CD3^+^IFNγ^+^/CD3^+^: WBM^DC-MLC(M)^: 55.25 ± 18.90; WBM^DC-MLC(Control)^: 38.98 ± 25.57, *p* = 0.008) (Figure 6, right side).

A parallel comparison of frequencies obtained from WB^DC-MLC^ vs. WBM^DC-MLC^ showed no significant differences (Figure 6, left side vs. right side).

The same applies to intracellularly TNFα-producing immune cells [37]. 

Kit-M-pretreated WB^DC-MLC(M)^ and WBM^DC-MLC(M)^ and untreated WB^DC-MLC(Control)^ and WBM^DC-MLC(Control)^ were intracellularly stained with an antibody against IFNγ to detect cytokine-secreting T cells. Given are the mean frequencies ± SD of T cell subsets secreting IFNγ in cultured leukemic WB (left side) and leukemic WBM (right side) without LAA stimulation. Differences were considered as “highly significant” in cases with *p*-values ≤ 0.005, as “significant” with *p*-values ≤ 0.05, as “borderline significant” with *p*-values between 0.05 and 0.1, and as “not significant” (n.s.) with *p*-values ≥ 0.1. Abbreviations of cell subpopulations are given in table in Section 4.3.

### 2.6. Effects of Kit-M-Treated WB and WBM on the Anti-Leukemic, Blastolytic Functionality after MLC

In WB, after 3 h, we observed blast lysis in 69% of Kit-M-treated cases (WB^DC-MLC(M)^) and in 62% of untreated cases (WB^DC-MLC(Control)^). After 24 h, blast lysis was achieved in all cases of WB^DC-MLC(M)^, whereas in WB^DC-MLC(Control)^, in only 79% of cases (Figure 7A left side). After 3 h, average frequencies of blasts showed an increase in blasts in WB^DC-MLC(Control)^ and blast lysis in WB^DC-MLC(M)^. After 24 h, we found a highly significant decrease in frequencies of blasts in WB^DC-MLC(M)^ vs. WB^DC-MLC(Control)^ (Figure 7B left side). Concerning the lysis of blasts in WB^DC-MLC(M)^ in relation to WB^DC-MLC(Control)^, we could observe improved lysis in 85% of cases after 3 h and in 93% of cases after 24 h (Figure 7C left side). Frequencies of improved lysis were lower after 24 h as compared to after 3 h (Figure 7D). Overall, choosing the best lysis and the most-improved lysis after 3 h or 24 h showed the same results as after 24 h.

In WBM, after 3 h, we observed blast lysis in 58% of cases in WBM^DC-MLC(M)^ and in 67% of cases in WBM^DC-MLC(Control)^. After 24 h, blast lysis was achieved in 92% of cases in WBM^DC-MLC(M)^ and in WBM^DC-MLC(Control)^ in 69% of cases (Figure 7A right side). After 3 h, blasts increased and average frequencies of increased blasts were (n.s.) higher in WB^DC-MLC(M)^ as compared to WB^DC-MLC(Control)^. After 24 h, we found significantly lower frequencies of lysed blasts in WBM^DC-MLC(M)^ vs. WBM^DC-MLC(Control)^ (Figure 7B right side). Concerning the lysis of blasts in WBM^DC-MLC(M)^ in relation to WBM^DC-MLC(Control)^, we could observe improved lysis in 50% of cases after 3 h and in 92% of cases after 24 h (Figure 7C right side). After 3 h, the average frequencies did not show an improvement in blast lysis of WB^DC-MLC(M)^ in relation to WB^DC-MLC(Control)^. However, after 24 h, improved lysis could be seen. (Figure 7D). Analogous to WB, the best lysis and the most-improved lysis after 3 h or 24 h coincide with the results obtained after 24 h.

Comparing the improved lysis in WB with WBM showed a significant difference (*p* = 0.02) in frequencies after 3 h, but not after 24 h (Figure 7D).

After MLC, WB^DC-MLC(M),(Control)^ and WBM^DC-MLC(M),(Control)^ served as “effector cells” and were cocultured with blasts (“target cells”) for 3 h and 24 h to evaluate the lytic activity of Kit-M-Treated WB and WBM. Given are the proportions of cases with lysis (Figure 7A), the mean frequencies ± SD of increased or lysed blasts (Figure 7B), the proportion of cases with improved blast lysis (Figure 7C) and the mean frequencies ± SD of improved lysed blasts of Kit-M-Treated samples in relation to control (Figure 7D) after 3 h, 24 h and the best after 3 h or 24 h in WB (left side) and WBM (right side). Differences were considered as “highly significant” in cases with *p*-values ≤ 0.005, as “significant” with *p*-values ≤ 0.05, as “borderline significant” with *p*-values between 0.05 and 0.1 and as “not significant” (n.s.) with *p*-values ≥ 0.1. Abbreviations of cell subpopulations are given in table in Section 4.3.

### 2.7. Correlation of Results with Patients’ Allocation to First Diagnosis and to Relapse after SCT

In patients at first diagnosis (*n* = 10) as well as at relapse after SCT (*n* = 6) higher frequencies of mature DC (subtypes) could be generated in Kit-M pretreated vs. control in WB as well as in WBM without induction of blast proliferation (e.g., at first diagnosis: %DC_mat_/WB: WB^DC(M)^: 11.20 ± 5.17; WB^DC(Control)^: 8.29 ± 3.20, *p* = 0.14; at relapse after SCT: %DC_mat_/WB: WB^DC(M)^: 13.59 ± 9.77; WB^DC(Control)^: 7.23 ± 3.14, *p* = 0.15; at first diagnosis: %DC_mat_/WBM: WBM^DC(M)^: 11.49 ± 5.28; WBM^DC(Control)^: 9.15 ± 4.36, *p* = 0.08; at relapse after SCT: %DC_mat_/WBM: WBM^DC(M)^: 13.17 ± 2.47; WBM^DC(Control)^: 8.26 ± 3.72, *p* = 0.07).

After MLC no significant differences in frequencies of (CD107a^+^ and IFNγ^+^) immunoreactive cells were found comparing samples at first diagnosis vs. at relapse after SCT in WB and WBM (e.g., %T_non-naive_CD107a^+^/T_non-naive_ (first diagnosis): WB^DC-MLC(M)^: 55.00 ± 26.55; WB^DC-MLC(Control)^: 45.71 ± 24.13, *p* = 0.01; %T_non-naive_CD107a^+^/T_non-naive_ (relapse after SCT): WB^DC-MLC(M)^: 53.91 ± 24.76; WB^DC-MLC(Control)^: 49.76 ± 22.4, *p* = 0.04).

Improved lysis was not different between cases at first diagnosis and relapse after SCT in WB as well as in WBM.

### 2.8. Correlation of Antileukemic Cytotoxicity with Patients’ Allocation to Risk Groups

Cases with improved lysis after 3 h and 24 h and the best after 3 h or 24 h were allocated to patients’ favorable (*n* = 3) vs. adverse (*n* = 6) cytogenetic risk allocation.

In WB, 100% of cases with favorable risk and 83.33% of cases of adverse risk achieved improved lysis after 3 h and 24 h (Figure 8A, left side).

In WBM, 100% of cases with favorable risk achieved improved lysis after 3 h and 24 h. In cases of adverse risk improved lysis was achieved in 16.67% of cases after 3 h and in 83.33% of cases after 24 h (Figure 8A, right side).

### 2.9. Correlation of Antileukemic Cytotoxicity with Patients’ Allocation to Response to Induction Chemotherapy

Cases with improved lysis after 3 h, 24 h, and the best after 3 h or 24 h of coincubation were allocated to patients responding (“responders”) (*n* = 3) and non-responding (“non-responders”) (*n* = 5) to induction chemotherapy. In WB, 100% of responders and 80% of non-responders achieved improved lysis after 3 h and 24 h (Figure 8B, left side). In WBM, 100% of responders achieved improved lysis after 3 h and 24 h. Non-responders achieved improved lysis in 20% of cases after 3 h and in 80% of cases after 24 h (Figure 8B, right side).

WB (left side) and WBM (right side) samples were allocated to cytogenetic ELN risk groups (Figure 8A) and patients’ response to induction chemotherapy (Figure 8B). Given the proportions of cases with improved blast lysis after 3 h, 24 h and the best after 3 h or 24 h.

### 2.10. Effects of Kit-M-Treated WB and WBM on Platelet Counts

We found an average platelet count of 110.8 × 103/µL in WB (range: 59–215 × 103/µL) and of 82.8 × 103/µL in WBM (range: 35–225 × 103/µL). We studied the influence of Kit-M vs. GM-CSF alone on the provision of platelets in the course of culture in 5 cases: we found that in WB, on average platelet counts increased under the influence of GM-CSF alone as well as of GM-CSF+PGE-1 (Kit-M) (Figure 9). A detailed analysis showed that in two cases (1601, 1620) in WB results after GM-CSF vs. Kit-M treatment were comparable, leading to increased platelet counts, whereas in another case (1618), they decreased. In case 1621 platelets were more increased under the influence of GM-CSF vs. Kit-M, while in case 1624 Kit-M produced more platelets vs. GM-CSF.

In WBM, average platelet counts increased under the influence of GM-CSF alone as well as after Kit-M pretreatment. In two cases (1620, 1621), results of GM-CSF and Kit-M were comparable-leading to increased platelet counts, whereas in others (1624) they decreased. In cases 1601 and 1618 platelets decreased under GM-CSF treatment, whereas Kit-M treatment led to an increase.

Platelets were counted on day 0, 3, and 7 after culture of Kit-M-treated WB (Kit-M WB) and WBM (Kit-M WBM). GM-CSF-treated WB (GM-CSF WB) and WBM (GM-CSF WBM) served as a control. The average fold change values of platelets ± SD as well as single case fold changes are given in WB (Figure 9A) and WBM (Figure 9B).

## 3. Discussion

### 3.1. WBM-the Immunosuppressive Leukemic Microenvironment

The physiological WBM-microenvironment (consisting of osteoblasts, MSC, HSC, adipocytes etc.) enables a strictly regulated hematopoiesis in the steady state as well as under stress [6,7]. However, the same microenvironment seems crucial in the transformation into a leukemic microenvironment through defective signaling pathways in specific cell types (e.g., β-catenin in osteoblasts, Notch signaling in endothelial cells) [39,40,41]. Leukemic cells and LSC express the chemokine receptor CXCR4 [42], which binds CXCL12, present on many WBM stromal cells [43]. With this binding, LSC replace HSC from their niche and remodel the microenvironment to their benefit in order to proliferate and migrate (e.g., adipocytes’ fatty acids as a source for leukemic metabolism [13], neoangiogenesis in WBM induced by blasts secreting angiogenic growth factors [9]). The leukemic microenvironment is furthermore characterized by its ability to support (immune) cells to bypass the immune system through a wide variety of mechanisms (e.g., impairing T cell function by downregulation of MHC II, expression of inhibitory checkpoint receptors (e.g., PD-1), increasing frequencies of T_reg_, inhibiting NK cell function, production of immunoinhibitory cytokines such as transforming growth factor beta (TGF β) and IL-10) [10,19]. These conditions and other mechanisms, which still need to be discovered in detail (e.g., LSC’s niche protection, self-renewal capacity, phenotypic/epigenetic plasticity), favor the persistence of quiescent and therapy-resistant LSC and were found mainly responsible for relapse [2,44].

Although promising approaches targeting blasts of any kind (e.g., PD-1 Inhibitors, CAR T cells, …) are in the focus of research [17], the greatest challenge remains to find a targeted therapy that attacks specifically these quiescent LSCs, bypasses immune escape mechanisms, acts throughout the body and does not affect healthy tissue.

### 3.2. DC-Based Immunotherapy

We successfully showed in previous studies that blasts can be turned regularly into DC_leu_ treating leukemic PBMC or WB ex vivo with combinations of response modifiers like Kit-M (GM-CSF and PGE-1) [35]. DC/DC_leu_ are cells able to migrate to tissues in the whole body, present patient-specific blast antigens in a costimulatory manner to immune cells, and thus, induce potent patient-specific anti-leukemic activity [32,45]. To expand our understanding of the potential effects of Kit-M, in this study, we treated WBM with Kit-M and compared results to Kit-M-treated WB.

### 3.3. Composition of Uncultured WB and WBM

Since average blast counts were 20% in uncultured WB and 26% in uncultured WBM as detected by flow cytometry, achieved results in WB and WBM were comparable. Frequencies of uncultured and leukemia-specific (degranulating or IFNγ-producing) immunoreactive cells with or without LAA (WT-1 and PRAME) stimulation were comparable in WB vs. WBM (Figure 1 and Figure 2). Interestingly, we found significantly higher frequencies of IFNγ-producing T_cm_ in WBM^+LAA^ compared to WBM^−LAA^ (Figure 2). This data confirm that LAA stimulation can increase the provision of leukemia-specific immunoreactive cells [46]. Here, we could confirm these findings in some subtypes. Comparing uncultured WB^−LAA^ vs. WBM^−LAA^ significantly lower frequencies of degranulating T_reg_ were found in WB. This was consistent with previous studies suggesting WBM being the preferential site of migration and function of T_reg_ going along with a higher activity of suppressive cells [11].

### 3.4. Generation of (Mature) DC/DC_leu_ in Kit-M-Treated WB and WBM

We could confirm preliminary data and generated significantly higher frequencies of DC/DC_leu_ in Kit-M-treated WB vs. control [32,35,46]. Comparable results were obtained in WBM. By generating higher frequencies of DC_mat/leu-mat_ in both Kit-M-treated WB and Kit-M-treated WBM vs. control (Figure 3) we yield proof that in both compartments (WB and WBM) DC/DC_leu_ with migratory capacities (allowing their migration to lymph nodes and activating immune cells and allowing immunoreactive effects all over the body) were created [47]. Regarding safety considerations, we could confirm previous findings that blast proliferation was not induced in either WB or WBM [34].

### 3.5. Provision of Immunoreactive Cells after MLC in Kit-M-Treated WB and WBM

As previously shown, we could observe a generally higher activation status of immune cells comparing leukemic WB after a T cell-enriched MLC and before. Similar results were found in WBM. These results can probably be attributed to the effects of IL-2 stimulation [38,48,49]. In WB, Kit-M treatment led to significantly higher frequencies of (activated) T cell subsets (e.g., T_prol-early_, T_non-naive_) after MLC as compared to control (Figure 4). Here, we could confirm previous findings [48] and expand the potential effects of Kit-M to WBM, as comparable results were obtained. In the past, reduced β7-expressing T cells in leukemic compared to healthy samples were found, resembling an effect of leukemia-associated immunosuppression [50]. We showed increased β7-expressing T cells in Kit-M-treated leukemic WB and WBM as compared to control. This was previously correlated with a higher antileukemic potential in WB samples [46] and could now be confirmed also for WBM. High frequencies of regulatory cells (e.g., T_reg_ and CD152 expressing T cells) could already be correlated with immunosuppressive effects in leukemia [48,51,52]. In both Kit-M-treated WB and WBM, we were able to show decreased frequencies of these subtypes compared to control, suggesting a transformation into a less immunosuppressive environment. Comparing frequencies of WB’s and WBM’s control groups after MLC, we found significantly higher frequencies of CD3^+^CD152^+^ cells in WBM, confirming findings described in Section 3.3 about WBM being the preferential site of immunosuppression mediated by T_reg_ [11]. Innate immune cells, like NK, CIK, and iNKT cells, were shown to crosstalk with DC and lead to antitumor reactions [53,54]. We could already show a (significant) increase in NK, CIK, and iNKT cells in Kit-M-treated WB vs. control (e.g., [53]).

As already shown, we could demonstrate that in cultures with patients’ cells at relapse after SCT (characterized by different chimerism), comparable antileukemic effects could be achieved, indicating a general blast modulatory/immune-activating mechanism in samples before or after SCT [55]. Here, we confirm these findings also for WBM, supporting the hypothesis that Kit-M treatment can contribute to overcome leukemic WBM’s immunosuppressive environment, although these findings might not be true for iNKT cells in WBM (pointing to a more immunosuppressive environment in WBM vs. WB as shown before) [8,10].

### 3.6. Enhanced Degranulation Activity of T Cells after MLC in Kit-M-Treated WB and WBM

We could show an increased degranulation activity (CD107a^+^) of T cell (e.g., CD3^+^CD107a^+^, T_non-naive_CD107a^+^, T_em/eff_CD107a^+,^ and T_cm_CD107a^+^) subtypes, proving the induction of leukemia-specific effector and memory cells, as demonstrated before [34,46]. Moreover, we found decreased frequencies of degranulating T_reg_ in Kit-M-treated WB vs. control (Figure 5). We thereby confirm previous studies indicating that Kit-M treatment leads to an enhanced anti-leukemic response in WB [46,56]. In WBM, similar results were obtained. However, Kit-M treatment seemed not to reduce frequencies of degranulating T_reg_ in WBM, which might confirm previous findings that WBM resident T_reg_ might be characterized by more immunosuppressive effects than T_reg_ detected in WB [10,57].

To evaluate potential proleukemic effects of T_reg_’ composition, blastolytic activities must be analyzed (Section 3.8).

### 3.7. Enhanced IFNγ Production of T Cells after MLC in Kit-M-Treated WB and WBM

We could confirm preliminary data by showing increased IFNγ production of antileukemically active β7-expressing, non-naive, as well as memory T cells (probably responsible for long-lasting immunity) in Kit-M-treated WB vs. control [34]. Results obtained in WBM were comparable, thereby proving the induction of leukemia-specific effects also in WBM (Figure 6). This means that the decrease of IFNγ production (being an immunoinhibitory effect in uncultured leukemic WB and WBM) could be overcome, as an increase in active anti-leukemic cells was detected in both compartments [10].

### 3.8. Improved Blastolytic Activity after MLC in Kit-M-treated WB and WBM

We could demonstrate a (significantly) improved lytic activity against leukemic blasts in WB and WBM through Kit-M-mediated pretreatment (Figure 7). In WB, more cases achieved lysis and improved lysis after 24 h compared to after 3 h. In WBM, only 50% of cases achieved improved lysis after 3 h. However, after 24 h, the results were comparable to WB. This occurrence might be explained by different cytotoxic mechanisms induced in immunoreactive cells in WB vs. WBM: the early and fast-acting perforin-granzyme pathway and the late and slow-acting Fas/FasL pathway, which can run separately or synergistically [35,58].

Interestingly, after collocating patients to adverse and favorable risks, as well as to responders (to induction chemotherapy) and non-responders, cases with no improved lysis after 3 h could be correlated in the majority to WBM of patients with adverse risk profiles and non-responders (Figure 8). In these cases, the immunoinhibitory mechanisms might be more pronounced, thus taking longer for the immune system to bypass them and elicit an anti-leukemic response [59].

### 3.9. Correlation of Results with Patients’ Allocation to First Diagnosis and to Relapse after SCT

Comparison of results of Kit-M treatment (of WB and WBM) at first diagnosis vs. relapse after SCT showed no significant differences, suggesting potential effects of Kit-M treatment independent of patients’ disease stages, before or after SCT, thereby confirming data of Ugur et al [55].

### 3.10. Influence of Kit-M or GM-CSF on Platelet Counts in WB and WBM

Studying the influence of Kit-M or GM-CSF (without added PGE-1) on the provision of platelets in WB or WBM in five cases in the course of the culture, we found an average increase in platelets in WB (twofold) and in WBM (twofold) compared to the start of culture. This was true in the presence of GM-CSF alone or Kit-M, thereby pointing to a potential of GM-CSF to stimulate thrombocytopoiesis as shown before [60,61]. Interestingly, these effects were seen without adding thrombopoietin or other thrombocytopoiesis-stimulating factors, which might point to the presence of these factors in WB or WBM [62]. However, the induction of platelets was not observed in every given case, which may have been due to insufficient thrombocytopoiesis-stimulating factors in these cases. Moreover, Kit-M was shown to increase platelet counts compared to GM-CSF. This could point to an important potentially platelet-stimulating influence of Kit-M. These potential effects have to be explored in further (clinical) studies.

## 4. Materials and Methods

### 4.1. Sample Collection

Sample collection of heparinized WB and WBM was conducted after obtaining the written informed consent of the patients in accordance with the Declaration of Helsinki and the ethics committee of Ludwig-Maximilians-University-Hospital Munich (vote No. 339-05). Patient samples and clinical reports were provided by the University Hospital of Augsburg, the Diakonieklinikum Stuttgart, the Rotreuzklinikum Munich, and the St.-Josefs-Hospital Hagen.

### 4.2. Patients’ Characteristics

WB and WBM samples were collected from 17 AML patients with an average age of 62 (range 22–78) years and a female-to-male ratio of 1.8:1. The average blast frequencies (as detected by flow cytometry) were 20% in WB and 26% in WBM. Patients were categorized based on the French American British (FAB) and World Health Organization (WHO) classification, the cytogenetic risk stratification (in favorable, intermediate, and adverse risks according to the European Leukemia Network (ELN) guideline 2017, molecular mutations and numeral and structural chromosomal aberrations are given), the stage of the disease (first diagnosis, persisting disease, relapse after SCT) and the etiology (primary, secondary, or therapy-related AML). Moreover, blast counts and phenotypes, as evaluated by flow cytometry, in WB and WBM are given. In patients at first diagnosis, the response to induction chemotherapy, and in patients after SCT, the leukocyte chimerism are given. An overview is given in Table 1.

### 4.3. Flow Cytometry

To assess and quantify phenotypes, frequencies, subgroups and functionality of leukemic blasts, DCs, T cells, NK cells, CIK cells, and iNKT cells flow cytometric analyses before and after culture were performed. Abbreviations of all cell types are given in Table 2.

Panels with various monoclonal antibodies (moAbs) labelled with Fluorescein isothiocyanate (FITC), Phycoerythrin (PE), Phycoerythrin Cyanin 7 (PC7) or Allophycocyanin (APC) were selected for cell staining. Antibodies were provided by Beckman Coulter^a^ (Krefeld, Germany), Becton Dickinson^b^ (Heidelberg, Germany), BioLegend^c^ (Amsterdam, Netherlands), Miltenyi Biotec^d^ (Bergisch Gladbach, Germany), and Santa Cruz Biotechnology^e^ (Heidelberg, Germany).

Cells were stained with FITC-conjugated moAbs CD3^b^, CD4^b^, CD25^a^, CD33^a^, CD34^a^, CD71^a^, CD107^ac^, CD117^c^, CD154^b^, CD45RO^a^, IPO38^e^; PE-conjugated moAbs CD3^d^, CD4^b^, CD56^a^, CD80^a^, CD117^a^, CD127^a^, CD152^b^, CD206^a^, 6B11^b^, IFNγ^c^, TCRγδ^b^; PC7-conjugated moAbs CD3^a^, CD4^a^, CD25^b^, CD33^a^, CD34^a^, CD56^a^, CD117^a^, CD197^b^, TNFα^c^; and APC-conjugated moAbs CD3^a^, CD33^a^, CD34^a^, CD56^a^, CD80^c^, CD117^a^, CD137^b^, CD206^b^, CD45RO^c^, Integrinβ7^b^. Death cells were detected using 7AADb. Isotype controls were conducted according to the manufacturer’s instructions.

Prior to staining, erythrocytes in WB and WBM samples were lysed with lysing buffer (Becton Dickinson). Cells were then incubated with the corresponding moAbs for 15 min in the dark using a staining medium containing 95% PBS and 5% FCS (Biochrom, Berlin, Germany). Intracellular staining (IPO38, IFNγ, and TNFα) was conducted with the FIX&PERMTM Cell Fixation and Permeabilization Kit (ThermoFisher Scientific, Darmstadt, Germany).

All flow cytometric analyses were performed using the fluorescence-activated cell sorting flow cytometer FACSCalibur (Becton Dickinson) and the analysis software CellQuestPro 6.1 (Becton Dickinson) applying a refined gating strategy [34].

### 4.4. Sample Preparation

A fraction of WB and WBM was used directly to set up dendritic cell cultures and to perform the Degranulation Assay and the Intracellular Cytokine Assay. Peripheral blood mononuclear cells (PBMC) were isolated from WB using the Ficoll-Hypaque density gradient centrifugation (Biocoll separating solution, Biochrom, Berlin, Germany). Using the AML patients’ PBMC T cell separation was carried out with the MACS^®^ MicroBead Technology according to the manufacturer’s protocol (Miltenyi Biotec).

PBMC and T cells were frozen in a freezing medium with 70% RPMI-1640 (Biochrom) containing 100 U/mL penicillin and 0.1 mg/mL streptomycin (PAN-Biotech, Aidenbach, Germany) (RPMI/PS), 20% human serum (PAN-Biotech), and 10% dimethyl sulfoxide (Sigma Aldrich Chemie GmbH, Steinheim, Germany) and stored at −80 °C until use.

### 4.5. Dendritic Cell Culture

Dendritic cells (DC) and leukemia-derived dendritic cells (DC_leu_) were generated by treating WB and WBM in parallel with the DC/DC_leu_-generating protocol “Kit-M”, containing 800 U/mL Granulocyte-Macrophage-Colony-Stimulating-Factor (GM-CSF; Sanofi-Aventis, Frankfurt, Germany) and 1 µg/mL Prostaglandin-E1 (PGE1; Santa Cruz Biotechnology).

Therefore, WB and WBM were cultivated in sterile 24-well-plates (Cellstar®, Greiner Bio-One, Kremsmuenster, Austria) using 500 µL X-VivoTM 15 medium (Lonza, Verviers, Belgium) and 500µL WB and WBM. “Kit-M” was added on day 0 and after 2–3 days as described [35]. Untreated WB and WBM served as negative controls. DC cultures were incubated at 37 °C, 21% O_2_, and 10% CO_2_ for 7 days.

Flow cytometric analyses of DC-subtypes and proliferating blasts from both Kit-M-treated WB (WB^DC(M)^) and WBM (WBM^DC(M)^) and untreated WB (WB^DC(Control)^) and WBM (WBM^DC(Control)^) were performed before and after culture using a refined gating strategy [33,34]. In case of less than 1% blasts in the cell fractions, DC_leu_ and associated subgroups could not be evaluated.

### 4.6. T Cell-Enriched Mixed Lymphocyte Culture (MLC)

DC/DC_leu_ cultures were used as stimulators to activate T cell-enriched immune cells after MLC. Therefore, 2.5 × 105 cells from WB^DC(Control)/(M)^ and WBM^DC(Control)/(M)^ were cocultured in parallel with 1 × 106 previously thawed autologous T cells, diluted with RPMI/PS and incubated at 37 °C, 21% O_2_ and 10% CO_2_. 50 U/mL Interleukin-2 (IL-2, PeproTech, Berlin) was added to all cultures on day 0 and again after 2–3 and 5–6 days. Cells, later referred to as WB^DC-MLC(Control)/(M)^ and WBM^DC-MLC(Control)/(M)^, were harvested after 7 days.

Flow cytometric analyses of different immune cell subtypes were performed before and after culture [34].

### 4.7. Degranulation Assay (DEG)

As a marker for induced cell cytotoxicity, cell degranulation was quantified in uncultured WB and WBM and in WB^DC-MLC(Control)/(M)^ and WBM^DC-MLC(Control)/(M)^ using a FITC-conjugated antibody against CD107a. Uncultured WB and WBM were stimulated in parallel with leukemia-associated-antigens (LAA): 2 µg/mL “Wilms Tumor 1” (PepTivator^®^WT1, Miltenyi Biotec) and 2 µg/mL “Preferentially Expressed Antigen of Melanoma” (PepTivator^®^PRAME, Miltenyi Biotec). Unstimulated WB and WBM served as controls.

In previous studies, we showed that the addition of LAA after MLC had no further effect on immunoreactive cells [34]. Therefore, WB^DC-MLC(Control)/(M)^ and WBM^DC-MLC(Control)/(M)^ were not stimulated with WT1 and PRAME.

To avoid loss or weakening of CD107a antibodies’ fluorescence, 2 µg/mL Monensin solution (BioLegend) was added to the cultures. After an incubation of 16 h at 37 °C, 21% O_2_, and 10% CO_2_ cells were harvested, stained, and analyzed by flow cytometry [48].

### 4.8. Intracellular Cytokine Assay (INTCYT)

To quantify the intracellular production of Interferon-γ (IFNγ) and Tumor Necrosis Factor-α (TNFα) of different immune cells in uncultured WB and WBM and in WB^DC-MLC(Control)/(M)^ and WBM^DC-MLC(Control)/(M)^, the Intracellular Cytokine Assay was performed. In analogy to the DEG, only uncultured WB and WBM were stimulated with LAA. To avoid spontaneous cytokine secretion, 5 µg/mL Brefeldin A solution (BioLegend) was added. All cultures were incubated for 16 h at 37 °C, 21% O_2_, and 10% CO_2_. After harvest, intracellularly produced IFNγ and TNFα were evaluated (as described in Section 4.3) [34].

### 4.9. Cytotoxicity Fluorolysis Assay (CTX)

The Cytotoxicity Fluorolysis Assay was conducted to assess the lytic activity of T cell-enriched immunoreactive cells in WB^DC-MLC(Control)/(M)^ and WBM^DC-MLC(Control)/(M)^ (“effector cells”) against leukemic blasts (“target cells”). Therefore, effector and target cells (with a ratio of 1:1) were cocultured in a medium containing 85% RPMI/PS and 15% human serum and incubated for 3 and 24 h at 37 °C, 21% O_2_, and 10% CO_2_. Target cells were stained with respective antibodies before incubation. After harvest, 7AAD and a defined number of Fluorosphere beads (Beckman Coulter) were added. As a control, effector, and target cells were cultured separately and mixed shortly before measurements.

Flow cytometric analyses were performed after 3 and 24 h using a refined gating strategy [49]. The lytic activity against blasts (“blast lysis”) is defined as the difference in frequencies of viable blasts in the effector-target-cell-cultures as compared to controls, and “improved blast lysis” is defined as the difference in proportions of “blast lysis” achieved in WB^DC-MLC(M)^ as compared to WB^DC-MLC(Control)^ and in WBM^DC-MLC(M)^ as compared to WBM^DC-MLC(Control)^.

### 4.10. Platelet Counts

To assess the influence of Kit-M on platelet counts, Kit-M-pretreated WB and WBM were analyzed on day 0 and after 3 and 7 days of DC cultures using the Automated Hematology Analyser XP-300 (Sysmex, Norderstedt, Germany). GM-CSF-treated WB and WBM served as control groups.

### 4.11. Statistical Methods

Statistical analyses and figures were implemented with Excel 2022 (Microsoft, Redmond, WA, USA) and Prism 9 (GraphPad Software, San Diego, CA, USA). Data are presented as mean ± standard deviation (SD). Statistical comparisons between the two groups were performed using the two-tailed *t*-test. Differences were considered as “highly significant” in cases with *p*-values ≤ 0.005, as “significant” with *p*-values ≤ 0.05, as “borderline significant” with *p*-values between 0.05 and 0.1, and as “not significant” (n.s.) with *p*-values ≥ 0.1.

## 5. Conclusions

In this ex vivo study, we showed that DC_leu_-generation is possible in comparable frequencies in WB as well as in WBM with Kit-M (GM-CSF and PGE-1). Furthermore, we showed that Kit-M induces leukemia-specific/antileukemic innate immune cells and effector, as well as memory T cells in both WB and WBM. We showed increased antileukemic activity in WB and WBM. Studies are currently being planned to examine if these cells could qualify to overcome the inhibitory microenvironment in vivo, specifically eradicate leukemic cells and stabilize remissions in patients with leukemia before or after SCT.

## Figures and Tables

**Figure 1 ijms-24-17436-f001:**
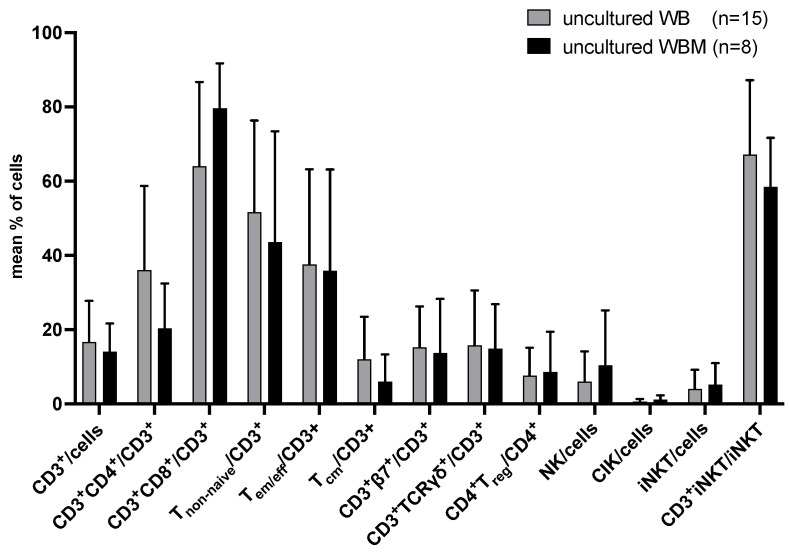
Composition of T and Innate Immune Cell Subtypes in Uncultured WB and WBM.

**Figure 2 ijms-24-17436-f002:**
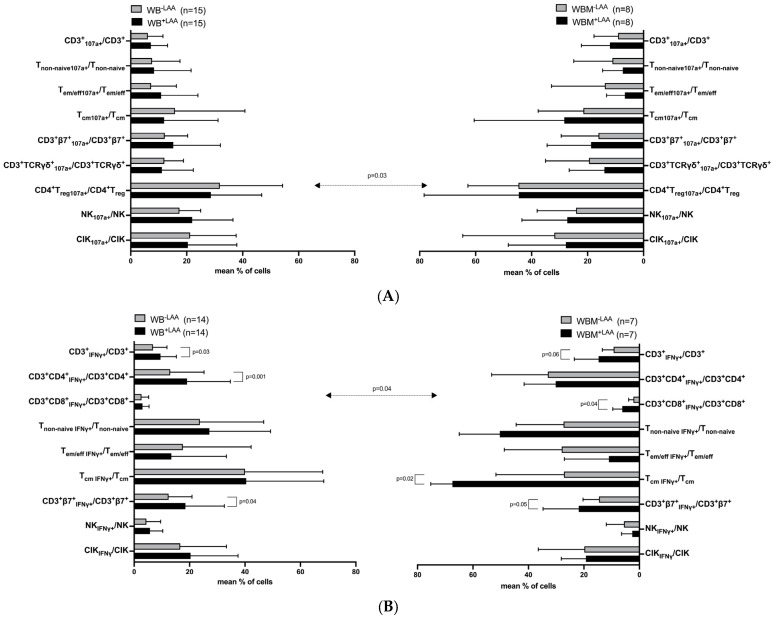
Composition of Leukemia-Specific Degranulating (**A**) or IFNγ-Secreting (**B**) Immunoreactive Cells in Uncultured WB and WBM.

**Figure 3 ijms-24-17436-f003:**
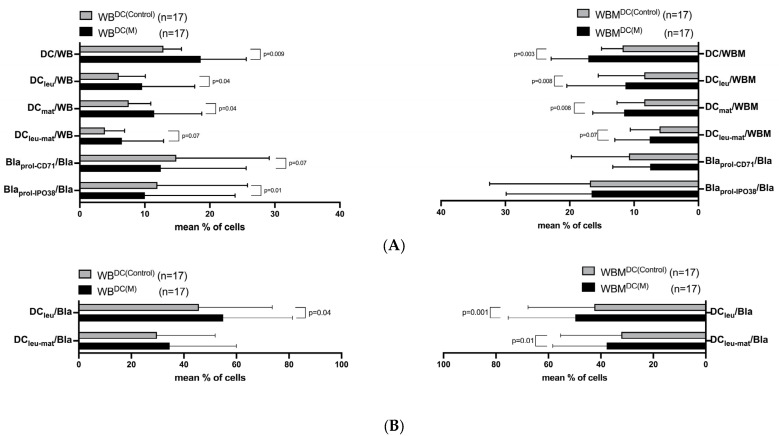
Effects of Kit-M on the Generation of (Mature) DC/DC_leu_ referred to WB (**A**) or Blasts (**B**) and on Blast Proliferation (**A**) in Leukemic WB and WBM.

**Figure 4 ijms-24-17436-f004:**
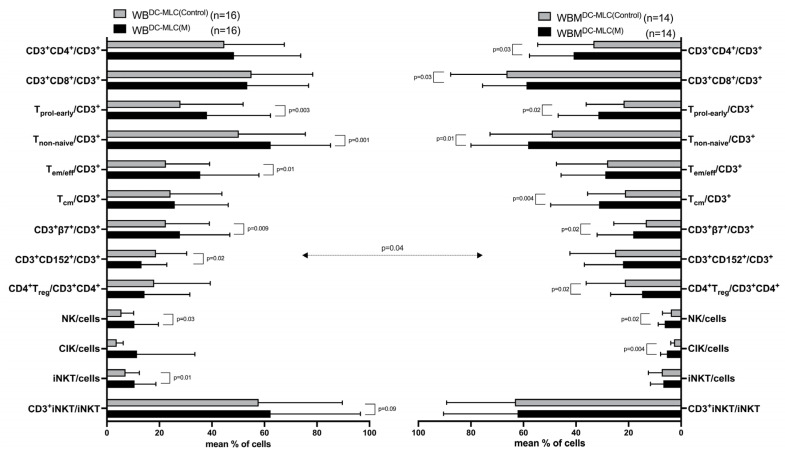
Effects of Kit-M-Treated WB and WBM on the Provision of Immunoreactive Cells After T cell-enriched Mixed Lymphocyte Culture (MLC).

**Figure 5 ijms-24-17436-f005:**
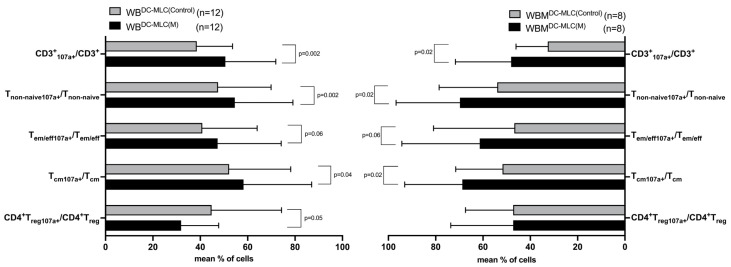
Effects of Kit-M-Treated WB and WBM on the Degranulation Activity of T cells After MLC.

**Figure 6 ijms-24-17436-f006:**
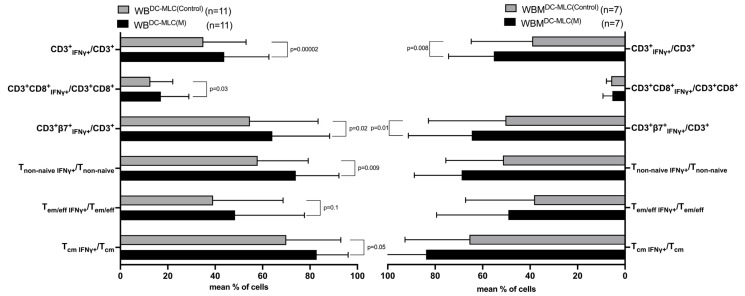
Effects of Kit-M-Treated WB and WBM on the Intracellular IFNγ Production of T cells After MLC.

**Figure 7 ijms-24-17436-f007:**
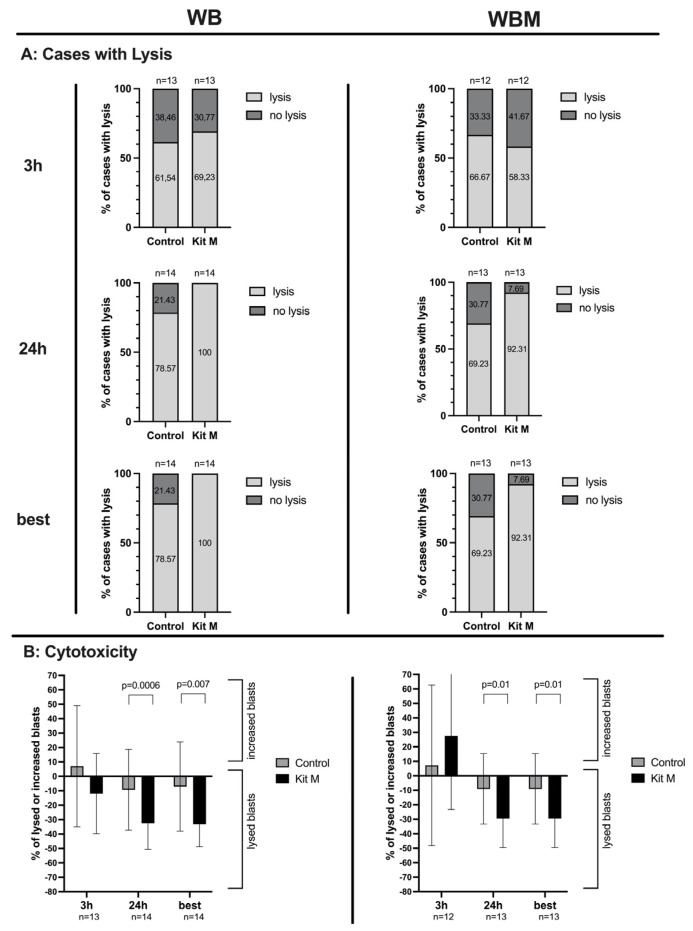
Effects of Kit-M-Treated WB and WBM on the Antileukemic Activity after MLC including Cases with Lysis (**A**), Cytotoxicity (**B**), Cases with Improved Lysis (**C**) and Improved Cytotoxicity (**D**).

**Figure 8 ijms-24-17436-f008:**
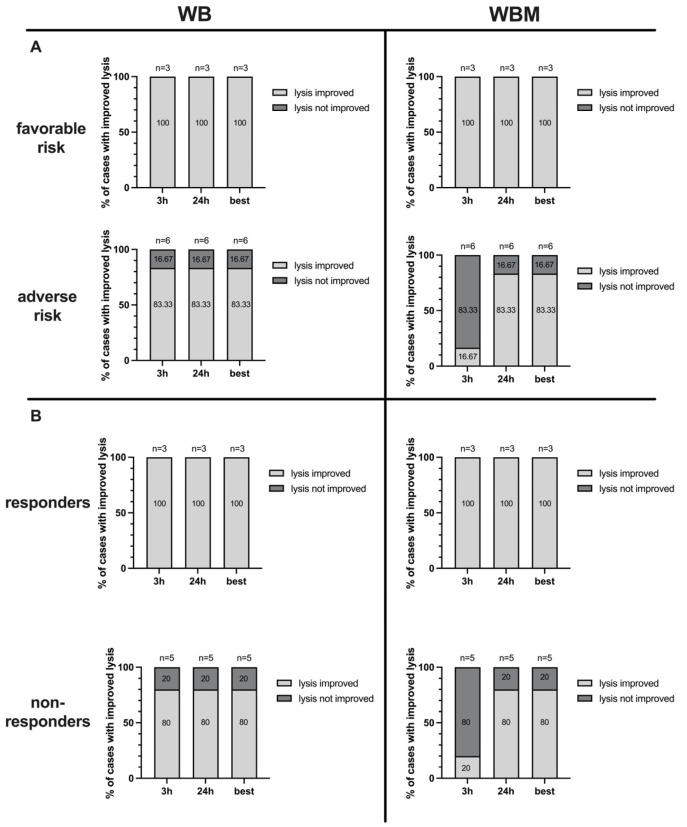
Cases with Improved Lysis in Patient Samples Subdivided into Risk Groups (**A**) and Response to Induction Chemotherapy (**B**) at First Diagnosis.

**Figure 9 ijms-24-17436-f009:**
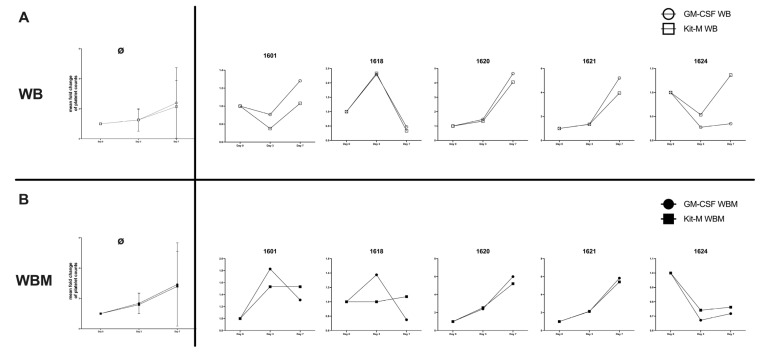
Effects of Kit-M or GM-CSF-Treated WB (**A**) and WBM (**B**) on Platelet Counts.

**Table 1 ijms-24-17436-t001:** Patients’ Characteristics.

Stage	Patient No.	Age at Sampling/Sex	FAB/WHO Classification	ELN (2017)-Risk-Stratification	Cyto-, Molecular Genetics	Blast Phenotype (CD)	IC in WB/WBM (%) *	Response to Induction Chemotherapy	Experiments Conducted with WB	Experiments Conducted with WBM
First Diagnosis	1615	78/F	sAML	Intermediate	*IDH1, ASXL1, JAK2, SRSF2*	**34**, **117**, 13	53/65	no	DC, MLC, Deg (UC, C), InCyt (UC, C), CTX	DC, MLC, CTX
	1618	63/M	pAML/M1	Favorable	*NPM1*	**117**, **33**, 13, 15	10/7	yes	DC, MLC, Deg (UC, C), InCyt (UC, C), CTX	DC, MLC, CTX
	1620	61/F	sAML	Adverse	t(2;3)(p23;q26), del (7)(q21), del(7)(q31), *MECOM* rearrangement, *ASXL1, RUNX1*	**34**, **117**, 33, 4	4/12	no	DC, MLC, Deg (UC), InCyt (UC), CTX	DC, MLC, CTX
	1621	71/M	sAML	Adverse	*TP53, ASXL1, SRSF2, SETPBP1, KRAS*	**34**, **117**, 5, 13	18/24	no	DC, MLC, Deg (UC, C), InCyt (UC, C), CTX	DC, MLC, CTX
	1624	77/F	sAML	Adverse	monosomy 7, der(7)t(1;7), der (2)t(2;3), del(7)(q31), ampl(3q26), *IDH2, STAG2, DNMT3A, MPL, NRAS, SRSF2*	**34**, **117**, 13	30/32	yes	DC, MLC, Deg (UC), InCyt (UC), CTX	DC, MLC, CTX
	1630	29/M	pAML	Favorable	*FLT3*-TKD, *NPM1, PTPN11*	**34**, **117**, 13, 33, 15, 64, 65	20/27	yes	DC, MLC, Deg (UC, C), InCyt (UC, C), CTX	DC, MLC, Deg (C), InCyt (C), CTX
	1638	68/M	sAML	Adverse	del(5)(q31), *TP53, RUNX1, RUNX1T1*	**34**, **117**, 33, 13, 56, 4	60/54	no	DC, MLC, Deg (UC, C), InCyt (UC, C), CTX	DC, MLC, Deg (UC, C), InCyt (UC, C), CTX
	1642	63/F	sAML	Adverse	*IDH2*, complex aberrant	**34**, **117**, 33, 13	1/19	yes	DC, MLC, Deg (UC, C), InCyt (UC, C), CTX	DC, MLC, Deg (UC, C), InCyt (UC, C), CTX
	1643	68/M	pAML	Favorable	*NPM1, IDH2*	**117**, **33**, 64, 13	23/39	no	DC, MLC, Deg (UC, C), InCyt (UC, C), CTX	DC, MLC, Deg (UC, C), InCyt (UC, C), CTX
	1645	58/F	pAML	Favorable	*IDH2, NPM1*	**117**, **33**, 56	36/47	yes	DC, MLC, Deg (UC), InCyt (UC)	DC, MLC, Deg (UC), InCyt (UC)
Persisting Disease	1601	75/F	sAML	Adverse	none detected	**34**, **117**, 33, 13	61/49		DC, MLC, Deg (UC), InCyt (UC), CTX	DC, MLC, CTX
								**Chimerism (%)**		
Relapse After Stem Cell Transplantation	1628	22/F	tAML/M5		*RUNX1, WT1, FLT3-ITD, KMT2A-PTD*	**34**, **117**, 65, 33, 56, 64	3/31	77	DC, Deg (UC), InCyt (UC)	DC
	1640	73/F	pAML		del(9q), *NMP1, IDH2, WT1, DNMT3A*	**34**, **117**, 33, 13, 7	<1/<1	56	DC, MLC, Deg (UC, C), InCyt (UC, C), CTX	DC, MLC, Deg (UC, C), InCyt (UC, C), CTX
	1641	64/F	tAML		dup(13)(q12q33), *DLEU* signal, *FLT3-ITD, EZH2, WT1*	**34**, **117**, 33, 13, 64, 65, 15	6/16	62	DC, MLC, Deg (UC, C), InCyt (UC, C), CTX	DC, MLC, Deg (UC, C), InCyt (UC, C), CTX
	1650	64/F	AML-MRC		*RUNX1, BCOR, KMT2A-PTD, TP53*	**34**, **117**, 13	3/8	86	DC, MLC, Deg (UC, C), InCyt (UC, C), CTX	DC, Deg (UC), InCyt (UC)
	1654	71/M	pAML/M2		der(16)t(16;17), *NPM1*	**117**, **56**, 13, 34	3/3	97	DC, MLC, Deg (UC, C), InCyt (UC, C), CTX	DC, MLC, Deg (UC, C), InCyt (UC, C), CTX
	1656	42/F	pAML		*MECOM* rearrangement, complex aberrant, *PTPN11*	**34**, **117**, 33, 56, 13	7/14	86	DC, MLC, Deg (UC, C), InCyt (UC, C), CTX	DC, MLC, Deg (C), InCyt (C), CTX

Legend. F: female; M: male; AML: acute myeloid leukemia; FAB classification: French American British classification of AML; WHO classification: World Health Organization classification of AML; pAML: primary AML; sAML: secondary AML; tAML: therapy-related AML; AML-MRC: AML with myelodysplasia-related changes; ELN: European Leukemia Network; del: deletion; t: translocation; der: derivative; ampl: amplification; dup: duplication; CD: Cluster of differentiation; bold: antibody used for expression analyses; WB: whole blood; WBM: WBM; no rsp: no response DC: dendritic cell culture measurements; MLC: mixed lymphocyte culture measurement; Deg: degranulation assay; InCyt: intracellular cytokine assay; CTX: cytotoxicity measurements; UC: measurements in uncultured samples; C: measurements after mixed lymphocyte culture; * Immune cytologically evaluated blasts.

**Table 2 ijms-24-17436-t002:** Cells and cell subsets as evaluated by flow cytometry.

Cell Type	Name of Subgroups	Abbreviation of Subgroups	Surface Marker	Referred to	Abbreviation	Reference
**Subtypes of blasts and DC**						
** *Blast cells* **	Leukemic blasts	Bla	Bla^+^ (e.g., CD34^+^, CD177^+^)	WB or WBM	Bla/WB or/WBM	[32]
	Proliferating blasts	Bla_prol-CD71_	Bla^+^DC^−^CD71^+^	Bla	Bla_prol-CD71_/Bla	[33]
	Proliferating blasts	Bla_prol-Ipo38_	Bla^+^DC^−^Ipo38^+^	Bla	Bla_prol-Ipo38_/Bla	[33]
** *Dendritic cells* **	Dendritic cells	DC	DC^+^ (CD80^+^, CD206^+^)	WB or WBM	DC/WB or/WBM	[35]
	Leukemia derived DC	DC_leu_	DC^+^Bla^+^	WB or WBM	DC_leu_/WB or/WBM	[35]
				Bla	DC_leu_/Bla	
	Mature DC	DC_mat_	DC^+^CD197^+^	WB or WBM	DC_mat_/WB or/WBM	[35]
	Mature DC_leu_	DC_leu-mat_	DC^+^CD197^+^Bla^+^	WB or WBM	DC_leu-mat_/WB or/WBM	[35]
				Bla	DC_leu-mat_/Bla	[35]
**Subtypes of Immune Reactive Cells**						
** *T cells* **	CD3^+^ pan T cells	CD3^+^	CD3^+^	cells	CD3^+^/cells	[36]
	CD4^+^coexpressing T cells	CD3^+^CD4^+^	CD3^+^CD4^+^	CD3^+^	CD3^+^CD4^+^/CD3^+^	[36]
	CD8^+^coexpressing T cells	CD3^+^CD8^+^	CD3^+^CD8^+^	CD3^+^	CD3^+^CD8^+^/CD3^+^	[36]
	Proliferating T cells-early	T_prol-early_	CD3^+^CD69+	CD3^+^	T_prol-early_/CD3^+^	[36]
	Non-naive T cells	T_non-naive_	CD3^+^CD45RO^+^	CD3^+^	T_non-naive_/CD3^+^	[34]
	Effector (memory) T cells	T_em/eff_	CD3^+^CD45RO^+^CD197^−^	CD3^+^	T_em/eff_/CD3^+^	[34]
	Central (memory) T cells	T_cm_	CD3^+^CD45RO^+^CD197^+^	CD3^+^	T_cm_/CD3^+^	[34]
	CD137^+^ coexpressing T cells	CD3^+^CD137^+^	CD3^+^CD137^+^	CD3^+^	CD3^+^CD137^+^/CD3^+^	[63]
	Integrinβ7^+^ coexpressing T cells	CD3^+^β7^+^	CD3^+^Integrinβ7^+^	CD3^+^	CD3^+^β7^+^/CD3^+^	[46]
	TCRγδ^+^ coexpressing T cells	CD3^+^TCRγδ	CD3^+^TCRγδ^+^	CD3^+^	CD^3+^TCRγδ^+^/CD3^+^	[46]
	CD152^+^ coexpressing T cells	CD3^+^CD152^+^	CD3^+^CD152^+^	CD3^+^	CD3^+^CD152^+^/CD3^+^	[52]
	Regulatory T cells	CD4^+^T_reg_	CD3^+^CD4^+^CD25^+^CD127^low^	CD3^+^CD4^+^	CD4^+^T_reg_/CD4^+^	[48]
** *Natural killer cells* **	CD3^−^CD56^+^NK cells	NK	CD3^−^CD56^+^	Cells	NK/cells	[53]
** *Cytokine induced killer cells* **	CD3^+^CD56^+^ CIK cells	CIK	CD3^+^CD56^+^	Cells	CIK/cells	[53]
** *Invariant natural killer T cells* **	6B11^+^ iNKT cells	iNKT	6B11^+^	Cells	iNKT/cells	[53]
	CD3^+^ coexpressing 6B11^+^ iNKT cells	CD3^+^iNKT	CD3^+^6B11^+^	6B11^+^	CD3^+^iNKT/iNKT	[53]
**Subtypes of different degranulating (CD107a^+^) cells as evaluated by flow cytometry**						
** *T cells* **	CD3^+^ pan T cells	CD3^+^_107a+_	CD107a^+^CD3^+^	CD3^+^	CD3^+^_107a+_/CD3^+^	[64]
	Non-naive T cells	T_non-naive107a+_	CD107a^+^CD3^+^CD45RO+	T_non-naive_	T_non-naive107a+_/T_non-naive_	[64]
	Effector (memory) T cells	T_em/eff107a+_	CD107a^+^CD3^+^CD45RO^+^CD197^−^	T_em/eff_	T_em/eff107a+_/T_em/eff_	[64]
	Central (memory) T cells	T_cm107a+_	CD107a^+^CD3^+^CD45RO^+^CD197^+^	T_cm_	T_cm107a+_/T_cm_	[64]
	Integrinβ7^+^ coexpressing T cells	CD3^+^β7^+^_107a+_	CD107a^+^CD3^+^Integrinβ7^+^	CD3^+^	CD3^+^β7^+^_107a+_/CD3^+^	[46]
	TCRγδ^+^ coexpressing T cells	CD3^+^TCRγδ^+^_107a+_	CD107a^+^CD3^+^TCRγδ^+^	CD3^+^TCRγδ^+^	CD3^+^TCRγδ^+^_107a+_/CD3^+^ TCRγδ^+^	[65]
	Regulatory T cells	CD4^+^T_reg107a+_	CD107a^+^CD4^+^CD25^+^CD127^low^	CD4^+^T_reg_	CD4^+^T_reg107a+_/CD4^+^T_reg_	[48]
** *NK cells* **	CD3^−^CD56^+^NK cells	NK_107a+_	CD107a^+^CD3^−^CD56^+^	NKcell	NK_107a+_/NK	[64]
** *CIK cells* **	CD3^+^CD56^+^ CIK cells	CIK_107a+_	CD107a^+^CD3^+^CD56^+^	CIKcell	CIK_107a+_/CIK	[64]
**Subtypes of different intracellularly IFNγ** or **TNFα-producing cells as evaluated by flow cytometry**						
** *T cells* **	CD3^+^ pan T cells	CD3^+^_IFNγ/TNFα_	IFNγ^+^/TNFα^+^CD3^+^	CD3^+^	CD3^+^_IFNγ/TNFα_/CD3^+^	[34]
	CD4^+^-coexpressing T cells	CD3^+^CD4^+^_IFNγ/TNFα_	IFNγ^+^/TNFα^+^CD3^+^CD4^+^	CD3^+^CD4^+^	CD3^+^CD4^+^_IFNγ/TNFα_/CD3^+^CD4^+^	[34]
	CD8^+^-coexpressing T cells	CD3^+^CD8^+^_IFNγ/TNFα_	IFNγ^+^/TNFα^+^CD3^+^CD8^+^	CD3^+^CD8^+^	CD3^+^CD4^−^_IFNγ/TNFα_/CD3^+^CD8^+^	[34]
	Non-naive T cells	T_non-naive IFNγ/TNFα_	IFNγ^+^/TNFα^+^CD3^+^CD45RO^+^	T_non-naive_	T_non-naive IFNγ/TNFα_/T_non-naive_	[34]
	Effector (memory) T cells	T_em/eff IFNγ/TNFα_/T_em/eff_	IFNγ^+^/TNFα^+^CD3^+^CD45RO^+^CD197^−^	T_em/eff_	T_em/eff IFNγ/TNFα_/T_em/eff_	[34]
	Central (memory) T cells	T_cm IFNγ/TNFα_	IFNγ^+^/TNFα^+^CD3^+^CD45RO^+^CD197^+^	T_cm_	T_cm IFNγ/TNFα_/T_cm_	[34]
	Integrinβ7^+^ coexpressing T cells	CD3^+^β7^+^_IFNγ/TNFα_	IFNγ^+^/TNFα^+^CD3^+^β7^+^	CD3^+^β7^+^	CD3^+^β7^+^_IFNγ/TNFα_/CD3^+^β7^+^	[46]
** *NK cells* **	CD3^−^CD56^+^NK cells	CD3^−^CD56^+^_IFNγ/TNFα_	IFNγ^+^/TNFα^+^CD3^−^CD56^+^	NK	NK_IFNγ/TNFα_/NK	[34]
** *CIK cells* **	CD3^+^CD56^+^ CIK cells	CI_IFNγ/TNFα_	IFNγ^+^/TNFα^+^CD3^+^CD56^+^	CIK	CIK_IFNγ/TNFα_/CIK	[34]

## Data Availability

The data presented in this study are available in this article.

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
