# Peer review of "Granulocyte-Macrophage-Colony-Stimulating-Factor Combined with Prostaglandin E1 Create Dendritic Cells of Leukemic Origin from AML Patients’ Whole Blood and Whole Bone Marrow That Mediate Antileukemic Processes after Mixed Lymphocyte Culture"

_ijms, 2023, doi:10.3390/ijms242417436_

Round 1

Reviewer 1 Report

Comments and Suggestions for Authors

The Authors present an interesting paper about an in vitro model of leukemic differentiation.

I would disagree with the first sentence of the abstract, given that in the ELN guidelines 2022 are provided several chemotherapy protocols for relapsed or refractory AML.

In the Introduction, Line 85, an 80% relapse rate after chemotherapy alone is mentioned. Where in the references is this rate? We have data in the literature that "favourable risk" AML has a DFS of 50-60% after 5 years. I am not confident that the 80% relapse rate is correct and should be amended.

Gemtuzumab Ozogamycin (there is a spelling error in the paper) is NOT a bispecific antibody, it's a toxin-bound antibody (Introduction, Line 89).

In the new therapies the IDH inhibitors Ivosidenib and Enasidenib should be mentioned.

Why is the materials and methods section after the discussion? It should be before the results.

What version of the ELN guidelines was used to classify the risk?

Table 1 is not clear: how many patients are at diagnosis, persisting disease (refractory?) or relapse after SCT?

In SCT patients, what was the chimerism? were the lymphocytes from the patient or from the donor?

The conclusions are way too broad. This is an invitro study, and only clinical in vivo research can tell of safety and efficacy.

Reviewer 2 Report

Comments and Suggestions for Authors

It’s an interesting study with lots of work, but I have following questions:

Please check your English, which has some mistakes. 

“Immune or immunoreactive cells” were used in the text, please make them more specific, T cell, or B cells, etc. 

Please define the first appearance of abbreviations such as the “WB” (Line 22)?

The writing of sentences line48-50 claiming FLT3/TP53 defects as the onset reason of AML is misleading. 

What do you mean controlled hematopoiesis? and BM niche as the main site (line55) needs references to support such description.  

Please introduce DC in more details including their subsets, molecular profiles, activation and their distribution under physiological condition. 

Please provide important profiles of cytogenetic and molecular characterizations of each patient. 

Please describe whether DCleu cells became normal or still contained the abnormal cytogenetics/mutations. 

Please provide the in vitro or ex vivo evidence or explain whether Kit-M would induce macrophages, M1 or M2 and their percentage, based on your experiments. 

Please describe the source of autologous T cells in details, which were ex vivo expanded from AML bone marrow or peripheral blood, or directly collected from AML patients?

Unfortunately, T cells cannot be isolated from every patient. Have you examined the co-culture of your inducible DC with allogeneic T cells?

Considering your experience of investigating these DCleu cells, please describe their engraftment in vivo including distribution. Can they migrate to physiological bone marrow (BM), or penetrate the leukemic BM? More importantly, did you observe any AML relapse or side effects after DCleu transplantation in vivo?

Have you evaluated the efficacy of Kit-M injection in vivo of AML mice? 

Comments on the Quality of English Language

Please check / correct some English mistakes.

Round 2

Reviewer 1 Report

Comments and Suggestions for Authors

The Authors have replied to my comments to my satisfaction, I have no further comments.